# Practical Application of Nanotechnology Solutions in Pavement Engineering: Identifying, Resolving and Preventing the Cause and Mechanism of Observed Distress Encountered in Practice during Construction Using Marginal Materials Stabilised with New-Age (Nano) Modified Emulsions (NME)

**Gerrit J. Jordaan** [1,2,*] and **Wynand J. vdM. Steyn** [3]

1 Department of Civil Engineering, University of Pretoria, Pretoria 0002, South Africa
2 Jordaan Professional Services (Pty) Ltd., Pretoria 0062, South Africa
3 School of Engineering and Department of Civil Engineering, University of Pretoria, Pretoria 0002, South Africa; wynand.steyn@up.ac.za
* Correspondence: jordaangj@tshepega.co.za; Tel.: +27-0-824164945

**Featured Application:** New-age (Nano) Modified Emulsions (NME) are applicable for the construction of high-order multi-lane highways to lower-order access roads in villages/townships. High-quality material-compatible anionic NME stabilising agents are user-friendly and forgiving in practice if basic normal construction controls and management procedures are in place. However, road construction projects are never without problems. New technologies that are introduced are easy targets to focus on, should any unrelated problems be experienced on site. Due to the lack of practical experience, it is important to correctly recognise, identify, and resolve the causes and mechanisms of observed distress during construction that are related to material and/or non-material related problems.

**Abstract:** New-age (Nano) Modified Emulsions (NME) for stabilising marginal materials used in the upper-pavement layers of roads have been proven in laboratories, through accelerated pavement tests (APT) in the field as well as in practice. In addition, materials design methods have been developed based on the scientific analysis of granular material mineralogy and the chemical interaction with the binder to design a material-compatible anionic NME stabilising agent for naturally available (often marginal) materials. However, any new disruptive technology that is introduced into a traditionally well-established industry, such as the road construction industry, is usually associated with considerable resistance. This is especially relevant when the new technology enables the use of granular materials traditionally considered to be of an unacceptable quality in combination with relatively new concepts such as an anionic NME stabilising agent. In practice, few road construction projects are without problems. New technologies are obviously easy targets to blame for any non-related problems that may arise during construction. In this article, we aim to assist in pre-empting, recognising, preventing, and resolving material or non-material related construction problems by correctly identifying the cause of the problems and recommending the best, most cost-effective ways to correct any deficiencies on site.

**Keywords:** new-age (Nano) modified emulsions (NME) stabilisation; identifying construction problems; preventing construction-related problems; material-related problems; constructability using nanotechnology applications; nano-silane stabilisation of granular materials; construction quality control problems; construction equipment problems; practical implementation of nano-silane stabilisation

## 1. Introduction

In the built environment, nano-silane products have been used in Europe since the 1800s [1–3] for protecting stone buildings against harsh environmental effects. These products have proven to be especially successful for protecting against water damage. Initial contradictory results soon enabled scientists to identify that the successful application of a specific product was a function of the type of stone and the condition of the stone [3]. In scientific terms, the type of stone is related to the primary minerals comprising the stone, while the condition of the stone is a function of the presence of secondary minerals as a result of weathering due to chemical decomposition.

Many buildings that have been successfully treated by scientists since the 1800s are still in everyday use, more than 150 years after a first treatment (e.g., silica-ester was recommended for treating the British parliament buildings in 1861 [2]). The same basic concepts that scientists developed by trial-and-error in the 1800s have been used in the modern era (after development of the advanced instruments in the 1980s/1990s that enabled scientists to manipulate atoms at a nano level, for example, atomic force microscopes (AFM) [4]), to develop nano-silane products that are now generally used in the built environment in numerous products, including silicon sealants, adhesives, and paints.

In the first decade of the 3rd millennium, the potential of nanotechnology applications was recognised in the field of road pavement engineering [5]. Most efforts have focused on improving bituminous surfacing, with numerous papers and articles published over the last decade [6,7]. However, the potential use of available, applicable and proven nanotechnology solutions to enhance and stabilise naturally available materials for use in the base and sub-base layers of roads (from local access roads to modern freeways) has received little attention [8]. These materials, in abundance in the developing world, are traditionally considered to be marginal, substandard, or even unsuitable [9] for use in these layers, based on traditional test and material characterisation methods for granular materials. It has been realised that the successful use of nanotechnology solutions to enable the use of these materials, without compromising the integrity of the pavement structure, could substantially reduce the unit costs of pavement structures [10]. Such developments could be of immense value to assist the developing world to build sustainable road infrastructures required to support economic development.

Over the last few years, the ability of these available nanotechnology solutions to stabilise, enhance, and improve naturally available materials has been proven in laboratories [11–13], through accelerated pavement testing (APT) [14–16], and in practice in southern Africa [8,17,18]. In parallel, to move away for the stigma of "snake oils" or "wonder products" [8,10,19], scientifically based design methods, that incorporate the basic scientific findings discovered in the 1800s (i.e., the type and condition of the stone) have been developed [19–23] to address any negative connections and engineering concerns. These methods are based on the basic mineralogy [19,20,23] of the naturally available materials and the chemical compatibility of the modified stabilising materials [22,23]. Adhering to fundamental principles [21], it is ensured that all nano-scale modifications to binders are safe both to the living as well as the environment. Due to the scientific basis of the materials design methods, no trial-and-error process or "proof of concept", so familiar in pavement engineering, is required if the basic materials design method [23] is followed and the required laboratory tests are performed to optimise designs in terms of fundamental engineering properties [19–23] and basic scientific principles.

Research on New-age (Nano) Modified Emulsions (NME) stabilising agents (including, but not limited to, nano-modified bitumen emulsions [8,11–23]) will ensure that engineers can perform scientifically based materials designs for road construction to limit risk by increasing aggregate adhesion and to provide sustainable hydrophobic (water repellent) pavement layers with confidence. The general introduction of these nanotechnologies could form the cornerstone of sustainable road networks with proven resistance to the:

- Destructive effect of water by introducing the concept of hydrophobicity to every particle within a pavement layer that is being treated to prevent future in-situ chemical decomposition [8];
- Formation of deep potholes when surfacing has been compromised [23]. and
- Destructive effect of overloaded heavy vehicles with damage factors (n) [8,15] below n = 2 (F = $(P/80)^n$), where P is the measured dual wheel single axle load measured in kN and F is the relative impact of a load (P) to a standard 80 kN dual wheel single axle load. It follows that the higher the damage factor (n) the greater the impact of axle load exceeding the standard of 80 kN will be on the pavement structure.

However, the introduction of any new disruptive technology [24], in a traditionally well-established industry such as the road construction industry, is usually associated with considerable resistance. This is especially relevant when the new technology is based on the use of:

- Granular materials traditionally considered to be of marginal or even unacceptable quality for use in specific pavement layers [9];
- Relatively new concepts in pavement engineering such as New-age (Nano) Modified Emulsions (NME) [8,10];
- Test requirements such as XRD scans [4,10] (relatively old concepts in fields such as geology and mining) to analyse naturally available granular materials [20], and
- Scientifically founded materials design methods based on the mineralogy of materials and material-compatible nano-modified stabilising agents [19,23] and the use of material tests indicative of fundamental engineering properties, such as the Unconfined Compressive Strength (UCS) [25] and Indirect Tensile Strength (ITS) [26].

Although most of these concepts have been in used for many a decade, if not for more than a century in the built environment [1–3], the traditional road construction industry is notoriously conservative. In terms of the use of material-compatible nanotechnologies, benefits have been demonstrated for construction methods ranging from the most basic to the most advanced [18]. However, it is rare that construction projects are conducted without any problems. In the case of relatively new materials (nanotechnology modifications), this give contractors an obvious scapegoat to blame in view of any identified construction problems, especially in the presence of supervision personnel that are not very experienced.

The main objective of this article is aimed at pre-empting the "blame game" during construction, by giving supervision engineers insight into typical problems that may be experienced on site. This is done by:

- Identifying the most prominent causes of NME technology that result in stabilised layers not meeting specifications, which can easily be prevented by following sound construction processes (provided that developed materials design methods are followed [8,23]), using material compatible modified stabilising agents, and
- Showing and discussing typical examples of problems encountered during the implementation of anionic NME technology using anionic modified bitumen emulsion stabilising agents on ±10 actual road construction projects and, in the case of material-compatible products resulting in construction-related problems, drawn from experience on countless examples in practice.

Initially, all the problems shown were immediately blamed on the new NME technology together with the use of "unsuitable" naturally available materials to construct the pavement layers. However, after thorough investigations (some resulting even in forensic investigations), all the problems shown were identified as typical construction-related (equipment and/or procedural) problems. In these cases, low-risk, cost-effective solutions could be recommended to resolve these problems on site.

None of the problems shown and discussed in this article were due to the incorporation of new anionic NME stabilising agents enabling the use of "unsuitable" naturally available granular materials. However, in practice, on a construction site, any new technology that is introduced is based on the concept of "guilty until proven innocent". Hence, field

experience using new technologies needs to be documented in order to pre-empt and limit opportunities for potential unsubstantiated claims.

## 2. Design and Investigations Preceding Construction

The construction of any road should be preceded by the usual design method prescribed by any specific road authority. These design methods and especially the methods of materials investigation, may be a function of new road design (e.g., [27]), upgrading and/or rehabilitation of existing surfaced roads (e.g., [28]), or the upgrading of gravel roads (e.g., [29]). With the implementation of nanotechnology solutions for stabilising and utilising marginal naturally available materials, materials tests will be included to provide the scientific information required to design material-compatible anionic NME stabilising agents [19–23].

It follows that before construction commences, available materials in borrow pits and/or in situ materials within existing roads (for upgrading or rehabilitation) must be sampled and tested. The testing, if done according to recommended best practice, ensures that the data are accurate, adequate, and statistically meaningful [28,30,31]. Laboratory tests will already have been done to confirm that the available granular materials can successfully be stabilised using NME stabilising agents, meeting the specified requirements of the material classification as per Figure 1 [2,23] and Figure 2 [8,23,29] for the different pavement layers. Figures 1 and 2 only contain the required material properties to be tested and the material criteria to be met for the different material classes (from NME1 to NME 4 and NME4 to NME4-WC). The use of these material tests to design a material-compatible modifying agent based on the minerology of a material and the chemical interaction with the modifying agent, is fully addressed in the materials design method [23].

| Test or Indicator | Material[1] | Material classification | | | |
|---|---|---|---|---|---|
| | | NME1 | NME2 | NME3 | NME4 |
| **Minimum material requirements before stabilisation and/or treatment (Natural materials)** | | | | | |
| Material spec.(minimum) Unestablished material: Soaked CBR[2] (%) (Mod AASHTO) | NG /(CS) | > 45[2] (95%) ACV < 30% | > 25[2] (95%) | > 10[2] (93%) | > 7[2] (93%) |
| Grading Modulus (GM) | NG | > 1.8 | > 1.5 | - | - |
| | GS | NA | > 1.5 | - | - |
| Sieve analysis: % < 0.075 mm (P0.075) | ALL | < 20% | < 25 % | < 35 % | < 50 % |
| XRD scans: - Total sample - 0.075 mm fraction (P0.075) | ALL ALL | √ √ | √ √ | √ √ | √ √ |
| % Material passing 2 µm (P0.002) (e.g. Clay & Mica & Talc) as a % of Material (with Talc <10%) (XRD-scans of the material passing the 0.075 mm sieve is used to determine the % clay, mica (muscovite) and talc in the material – In this case P0.002 = P0.075 x (Pclay, etc in P0.075) | ALL | NME stabilisation with micro-meter (µm) emulsion particle sizes | | | |
| | | < 15 % | < 15 % | < 15 % | < 15 % |
| | ALL | NME stabilisation with emulsion containing micro-scale as well as nano-scale particles (adjusted according to material grading) | | | |
| | | NA | < 35 % | < 35 % | < 35 % |
| | ALL | NME stabilisation with emulsion containing nano-scale and pico-scale particles (grading adjustments) together with technologies addressing workability of materials on site | | | |
| | | NA | NA | > 35 % | > 35% |
| **Material specifications after stabilisation and/or treatment** | | | | | |
| In-situ density to be required after stabilisation and compaction (mod AASHTO) (%) (minimum) | Base | > 100 % | > 100 % | > 98 % | > 97 % |
| | Sub-base | NA | > 98 % | > 97 % | > 95 % |
| DCP(DN mm/blow)(Quality control) (stabilised and compacted) | | NA | NA | < 2.6 | < 3.5 |
| Mod AASHTO density (%) (for laboratory testing) | | > 100 % | > 100 % | > 100 % | > 100 % |
| *UCSwet (kPa) (150 mm Φ Sample) | Design[3] | > 2 500 | > 1 500 | > 1 000 | > 750 |
| | Construction[4] | > 2 200 | > 1 200[5] | > 700[5] | > 450[5] |
| Retained Compressive Strength (RCS): (UCSwet/UCSdry) (%) | | > 85 | > 75 | > 70 | > 65 |
| RCS in relation to minimum UCSwet(criteria) = RCSeffective = (RCS x (UCSwet/UCSwet(criteria))) (%) | | >100 | >100 | >100 | > 100 |
| *ITSwet (kPa) (150 mm Φ Sample) | Design[3] | > 240 | > 200 | > 160 | > 120 |
| | Construction[4] | > 220 | > 180[5] | > 140[5] | > 100[5] |
| Retained Tensile strength (RTS): ITSwet/ITSdry (%) | | > 85 | > 75 | > 70 | > 65 |
| RTS in relation to minimum ITSwet(criteria) = RTSeffective = ((RTS x (ITSwet/ITSwet(criteria))) (%) | | >100 | >100 | >100 | > 100 |

[1]CS – crushed stone; NG – natural gravel; GS – gravel soil, and SSSC – sand, silty sand, silt, clay.\
[2]CBR only used as reference to traditionally used test procedures as a broad first indicator
*Definitions: UCS = Unconfined Compressive Strength; ITS = Indirect Tensile Strength);
　UCSdry; ITSdry = testing after rapid curing; UCSwet; ITSwet = testing after rapid curing and 4 hours in water (as per test procedure specified for the testing of cementitious stabilising agents (SANS 3001-GR32:2010, 2010));
Design[3] = Minimum criteria to be met in the laboratory during the design phase
Construction[4] = Minimum criteria to be met during construction as part of quality control
[5]Criateria based on reference TG2 (Asphalt Academy, 2009)

**Figure 1.** Minimum recommended standard specifications for New-age (Nano) Modified Emulsions (NME) stabilise materials, addressing four different classifications in terms of engineering requirements for primary (highways), secondary, and tertiary roads [8,23].

| | Material[1] | Material classification NME4 - Gravel roads | |
|---|---|---|---|
| **Minimum material requirements before stabilisation and/or treatment (Natural materials)** | | | |
| Material spec.(minimum) Unstabilised material: Soaked CBR (%)(Mod AASHTO) | NG/GS/SSSG (CS) | > 7 (93%) | |
| Sieve analysis % passing the 0.075 mm sieve ($P_{0.075}$) | | < 50 % | |
| XRD scans: <br> - Total sample <br> - 0.075 mm fraction | ALL <br> ALL | √ <br> √ | |
| The greater of: | | NME stabilisation with emulsion particle size > 2 µm | |
| | ALL | < 15 % | |
| Identified % Silt and Clay, or | | NME stabilisation with emulsion containing micro-scale as well as nano-scale particles (adjusted according to material grading) | |
| % Material passing the 2 µm ($P_{0.002}$) sieve size (e.g. Clay & Mica & Talc) (with Talc <10%) (XRD-scans of the material passing the 0.075 mm sieve is recommended to be used to determine the % clay, mica and talc in the material) ($P_{0.075}$ x $P_{0.002}$) | ALL | ≥ 15% and < 35% | |
| | | NME stabilisation with emulsion containing nano-scale and pico-scale particles (grading adjustments) together with technologies addressing workability of materials on site | |
| | ALL | > 35% | |
| **Material specifications after stabilisation and/or treatment** | | **NME4** | **NME4-WC** |
| In-situ density to be required after stabilisation and compaction (mod AASHTO) (%) (minimum) | Base-layer | > 97 % | > 97 % |
| DCP DN (mm/blow) – (stabilised and compacted) (Quality control) | Top of base | < 3.5 | < 3.5 |
| Mod AASHTO density (%) (for laboratory testing) | | > 100 % | > 100 % |
| *$UCS_{wet}$ (kPa) (150 mm Φ Sample) | Design[3] | > 750 | > 750 |
| | Construction[4] | > 450[5] | > 450[5] |
| Retained Compressive Strength (RCS): ($UCS_{wet}$/$UCS_{dry}$) (%) | | > 65 | > 60 |
| RCS in relation to minimum $UCS_{wet(criteria)}$ ($RCS_{effective}$): (RCS x ($UCS_{wet}$/$UCS_{wet(criteria)}$)) (%) | | > 100 | > 80 |
| *$ITS_{wet}$ (kPa) (150 mm Φ Sample) | Design[3] | > 120 | > 70 |
| | Construction[4] | > 100[5] | > 50[6] |
| Retained Tensile strength (RTS): $ITS_{wet}$/$ITS_{dry}$ (%) | | > 65 | > 50[6] |
| RTS in relation to minimum $ITS_{wet(criteria)}$ ($RTS_{effective}$) ((RTS x ($ITS_{wet}$/$ITS_{wet(criteria)}$)) (%) | | > 100 | > 80 |

CS – Crushed Stone; NG – Natural Gravel; GS – Gravel Soil, and SSSC - Sand, Silty sand, Silt, Clay.
[1]Definitions: $UCS_{dry}$; $ITS_{dry}$ = testing after rapid curing;
        $UCS_{wet}$; $ITS_{wet}$;= testing after rapid curing and 4 hours in water (as per test procedure specified for the testing of cementitious stabilising agents (SANS 3001-GR32:2010, 2010))
Design[3] = Minimum criteria to be met in the laboratory during the design phase
Construction[4] = Minimum criteria to be met during construction as part of quality control
[5]Criteria based on TG2 (Asphalt Academy, 2009); [6]Criteria (Jordaan et al, 2017b)

**Figure 2.** Minimum recommended standard specifications for New-age (Nano) Modified Emulsions (NME) stabilised materials, addressing low volume roads (LVR) and access roads to remote communities and in villages/townships, upgrading of gravel roads, and roads in most residential areas [8,23,29].

    Implementation of the materials design method [23] ensures that there is low or no possibility of the designed material-compatible NME stabilising agent being responsible for any construction-related problems. The risk is eliminated based on the implementation of the basic design requirements [21,23] and the selection of a material-compatible anionic NME stabilising agent based on recommended "end-product specifications" [8,29]. The "end product specifications" ensure that any NME product to be used, must be proven and guaranteed to meet the specified minimum requirements using the specific available natural granular materials. If all basic design procedures and tendering procedures have been fully implemented, verified, and approved, any problems experienced during construction, in all probability, will be related to the absence of experienced supervisory personnel and procedural- and/or equipment-related aspects.

### 3. Construction of NME Stabilised Granular Pavement Layers: The Most Prominent Basic Requirements to Prevent Construction Problems

#### 3.1. Construction Water Quality

    Similar to the construction of any pavement layer, when using any stabilising agent, the quality of construction water must meet the requirements generally contained in construction specifications (e.g., [8,29,30]). In rural areas, this can present some challenges, since water is often sourced from local streams without performing any water quality testing. Water is often sourced without adequate filtering to prevent small particles, etc., from being pumped into a water bowser. Basic preventative measures implemented in the sourcing of construction water can prevent many costly construction problems on site. A good rule of thumb is that construction water should also qualify for human consumption.

### 3.2. Clean Equipment

Thoroughly cleaned equipment is a prerequisite to successful application of an NME stabilising agent. Although contractors are aware of this aspect, almost without exception, problems occur on the first day of operations as a result of using equipment containing residue from previous operations. The organofunctional silane modification [22] of a stabilising agent (i.e., bitumen elusion or equivalent polymer) is a reactive agent that reacts with any bituminous residue left in a water bowser from previous operations. It has become the norm to expect reactions (as shown in Figure 3), questioning the quality of the stabilising agent when the NME stabilising agent is added to uncleaned water bowsers, resulting in the formation of "blobs" or sticky substances (Figure 3a,b) of bituminous materials that are unusable. In order to prevent such "first day" occurrences, it is recommended that equipment be inspected before being used on site for any NME stabilisation purposes. Figure 3c,d shows some results of pre-inspections done on "clean" equipment.

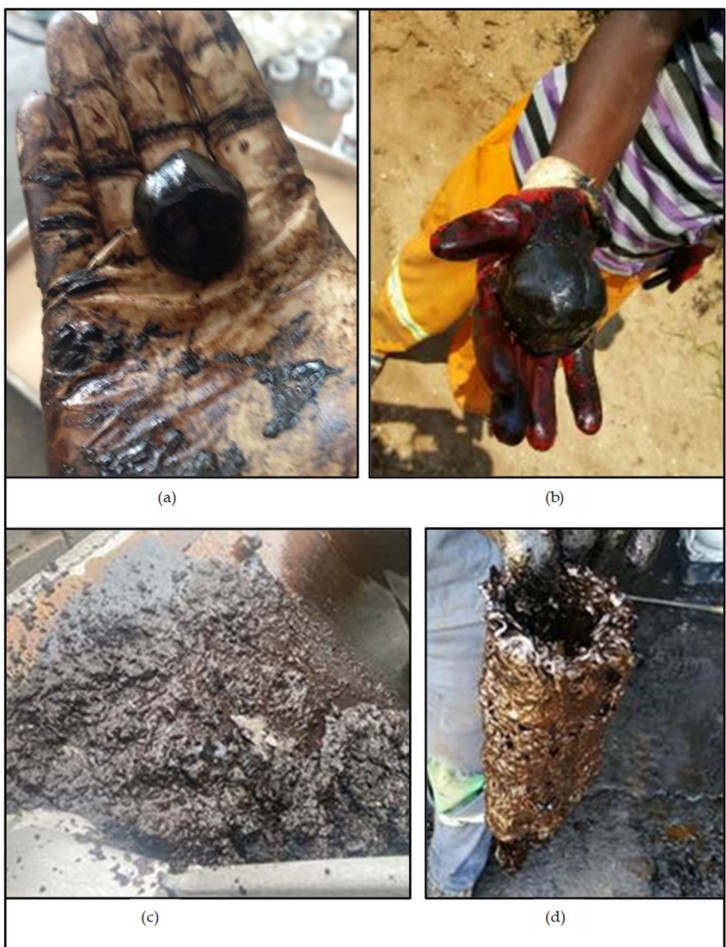

**Figure 3.** (**a**,**b**) Typical example of an anionic NME stabilising agent that has been added to a construction water bowser containing residue from previous operations and resulting in a reaction with the residue and an unusable "blob" of bituminous material; (**c**,**d**) typical examples of residue and the state of equipment found during the pre-inspection of "clean" equipment used by contractors for stabilisation of granular materials using an anionic NME.

## 4. Resolving Some Common Construction-Related Problems Experienced during the Stabilisation of Granular Materials Using New-Age (Nano) Modified Emulsions (NME)

### 4.1. Compaction of a NME Stabilised Granular Layer under Moisture Conditions That Are Too Low or Too High

In practice, compaction of a pavement layer under moisture conditions that are too low is easily recognised by observing the formation of small cracks that appear behind

a smooth-drum roller used for compaction. These cracks are often misdiagnosed on site as signs of "pumping", i.e., too high a moisture content, which requires drying out of the material before final compaction. Such actions only worsen the situation, basically ensuring that the layer will not meet the minimum specifications as per Figure 1 or Figure 2.

A typical example of such cracking is shown in Figure 4. It is always good practice to have a water bowser on stand-by with a small percentage of diluted NME in the construction water. As soon as cracking is noticed, a light spray with the diluted NME solution can provide the necessary moisture to compact the layer as a solid unit. The additional NME solution also enriches the surface of the layer providing additional strength to the top of the layer. As a rule of thumb, water without a stabilising agent should never be used to increase the moisture on surfacing, as it lacks the binder required to re-establish bonding of the materials already affected through the introduction of cracking.

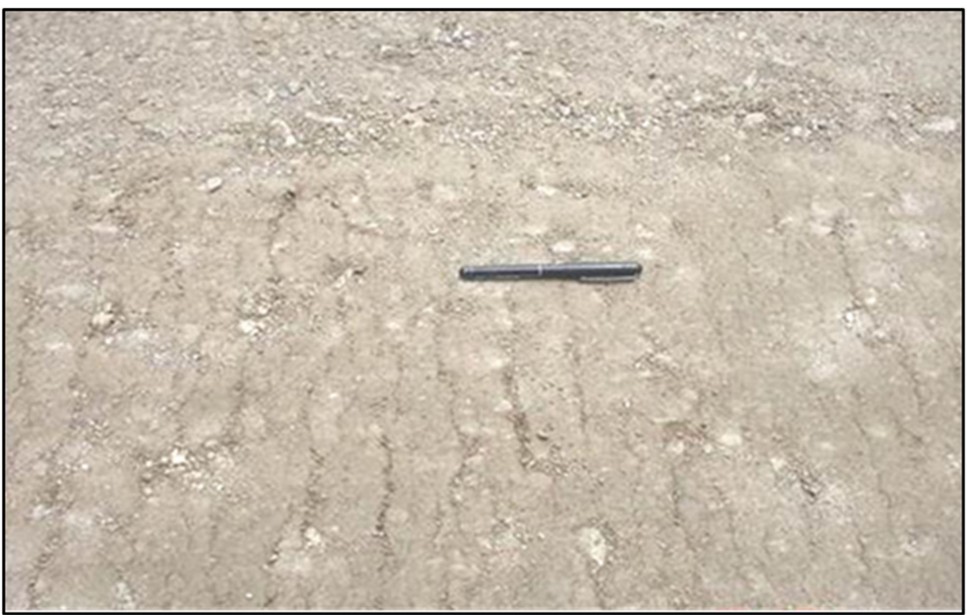

**Figure 4.** Appearance of small cracks on the surface directly behind a smooth-drum roller which are indicative of moisture conditions in the construction layer that are too low to achieve good compaction.

Experience has shown that stabilisation of granular materials using construction water diluted with a material-compatible NME stabilising agent are quite resilient and forgiving in nature. In contrast to compaction under moisture conditions that are too low, the opposite can also happen. In the case of moisture conditions that are too high, a small "wave" or deformation of material can be noticed to move in front of the smooth drum roller. In such cases, there is no damage caused by ripping the layer and allowing the material to be exposed to the sun for some of the moisture to evaporate. Re-compaction of the construction layer has little (if any) impact on the engineering strength requirements to be achieved.

In extreme cases, movement of the stabilised layer (even when walking on the layer) after some hours, can indicate stabilisation of a pavement layer to the wrong depth (too shallow), resulting in an overapplication of the stabilising agent in a thinner than specified layer. This operational problem is usually associated with inadequate training of equipment operators, be it with conventional equipment (grader) or a recycler. Localised opening and measuring of layer thicknesses should immediately confirm this aspect.

An extreme example of such an occurrence is shown in Figure 5. Closer investigations showed that stabilisation was confined to a depth of 50 mm only, instead of the specified 150 mm. The depth of stabilisation is the responsibility of the contractor who must ensure that specifications are met. In this case, the total rework of the layer to a depth of 150 mm is the obvious solution.

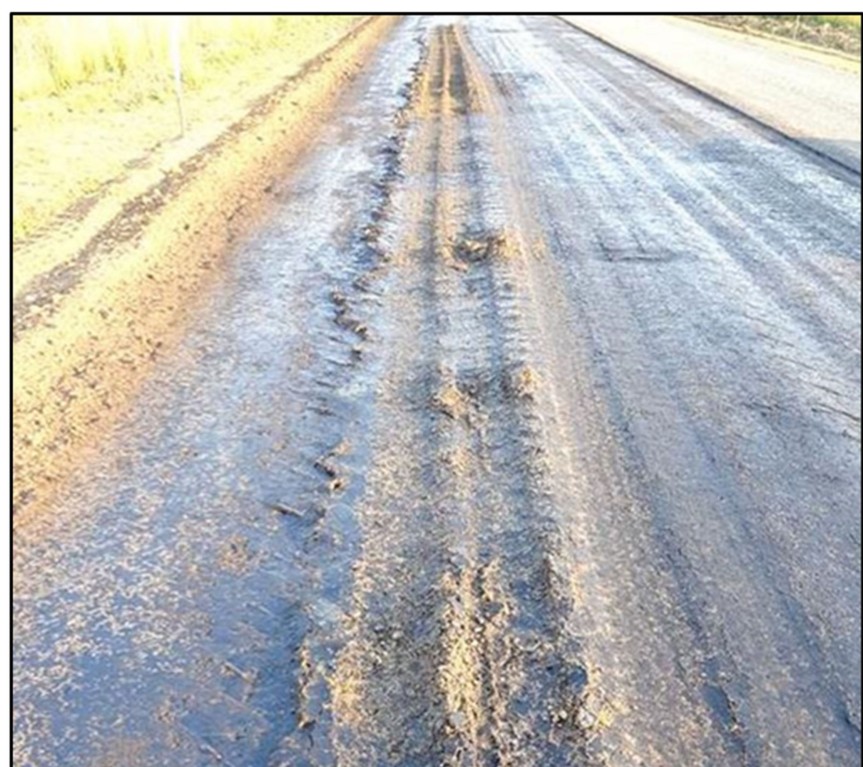

**Figure 5.** Severe movement of the NME stabilised layer, which is indicative of layer thicknesses that do not comply with specifications. The NME stabilising agent was added to a very thin layer (50 mm) instead of the specified 150 mm.

Due to the resilient nature of a material-compatible anionic NME stabilising agent, reworking can be achieved through in-situ milling, mixing and windrowing of 150 mm of material, followed by reinstatement of thoroughly mixed material and re-stabilisation by using 50 per cent of the original specified amount of NME stabilising agent. Care should be taken not to exceed the optimum moisture content (OMC) of the reinstated materials. The organofunctional silane modifying agent (NME) already added to the material reduced the original OMC of the material by at least 10 per cent. Hence, the best condition for compaction is a moisture content of about 10 per cent less than the original OMC to avoid movement of the material during re-compaction. Experienced supervision is key to recognising site conditions and making adjustments to the calculated recommended construction water to be used (taking into account practical site conditions such as the evaporation that may occur as a result of high temperature conditions during the processing of the material), to ensure that the material is not too dry and exhibiting cracking (as shown in Figure 4) or too wet, to immediately recognise such occasions, and to immediately address the problem on site.

*4.2. Reworking of Pavement Layers Not Meeting Criteria and Future Rehabilitation of NME Stabilised Layers*

4.2.1. Reworking of NME Stabilised Pavement Layers

Most construction contracts experience some challenges. In all probability, some NME stabilised sections may not meet the specified engineering criteria. There are numerous reasons for this problem, including equipment failure or a problem that results in a surfacing condition clearly not meeting requirements. Typical examples of construction equipment-related problems are shown in Figures 6 and 7.

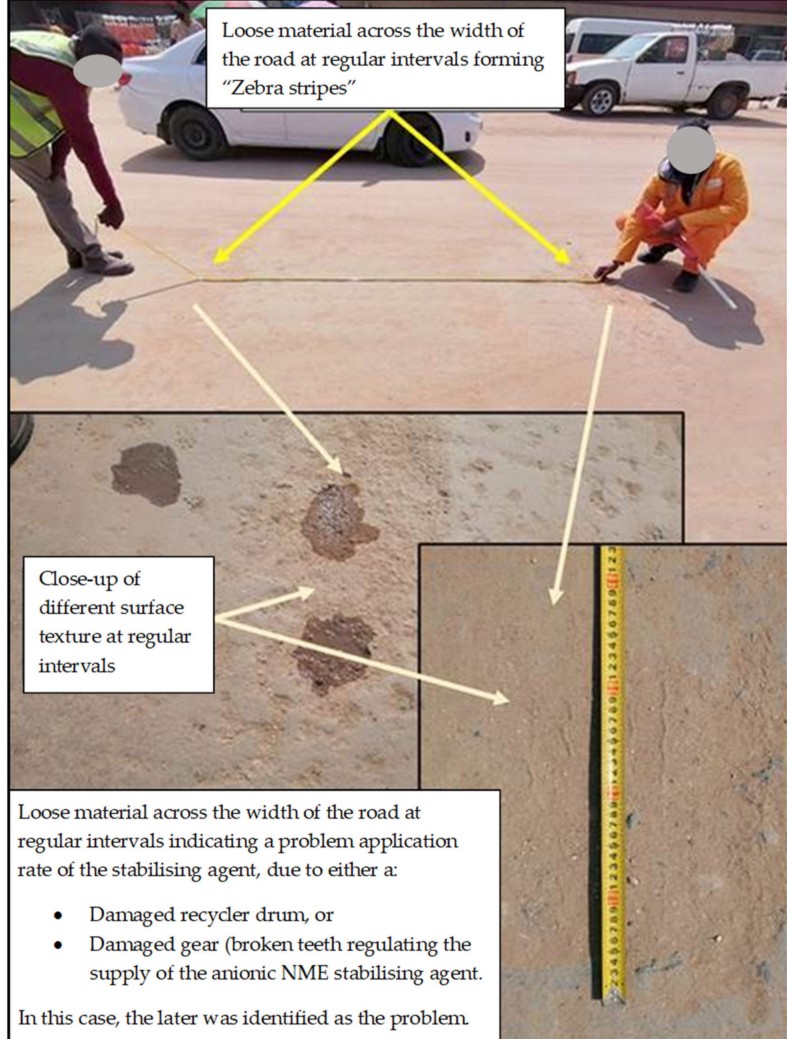

**Figure 6.** Construction equipment-related surfacing problem, showing surface irregularities at regular intervals across the width of the stabilised section. Stabilisation was done using recycling equipment (these problems are associated with the condition of the equipment used and not related to the stabilising agent).

As shown in Figure 6, the appearance of surface irregularities at regular intervals immediately rules out the possibility that the problems were associated with the stabilising agent. The spacing of the irregularities on the surface indicate that the problem is associated with the supply or mixing process and the equipment used in the distribution and mixing of the stabilising agent, i.e., in this case, a recycler. Two possible scenarios can be identified, i.e., either a drum-related problem or a problem associated with the supply of the stabilising agent within the recycler. In this case, a broken gear, regulating the supply of the diluted NME stabilising agent, resulted in a regular undersupply of the diluted NME stabilising agent which materialised on the pavement layer surface as shown.

The engineering specifications (NME3) for the specific layer (refer to Figure 1 or Figure 2) was exceeded by some margin (meeting the criteria of a NME1 material) throughout this constructed layer works, as shown in Table 1. At his own risk, the contractor reworked the specific section without any addition of stabilising agent or construction water within 24 h after initial construction. Although there was a reduction in the original quality control (UCS and ITS) measurements, the reworked road section still met the engineering requirements, the results of which are shown in Table 1. A decrease in the wet UCS and ITS values of less than 20 per cent was measured in this specific project. This example

is testimony to the resilient, forgiving nature of the stabilisation of naturally available gravel material (in this case G7 [27] quality material) stabilised with a material-compatible anionic NME stabilising agent. It should be emphasised that dry remixing of the layer is not the recommended method (as discussed in Section 4.1) to rework a layer not meeting the required specifications.

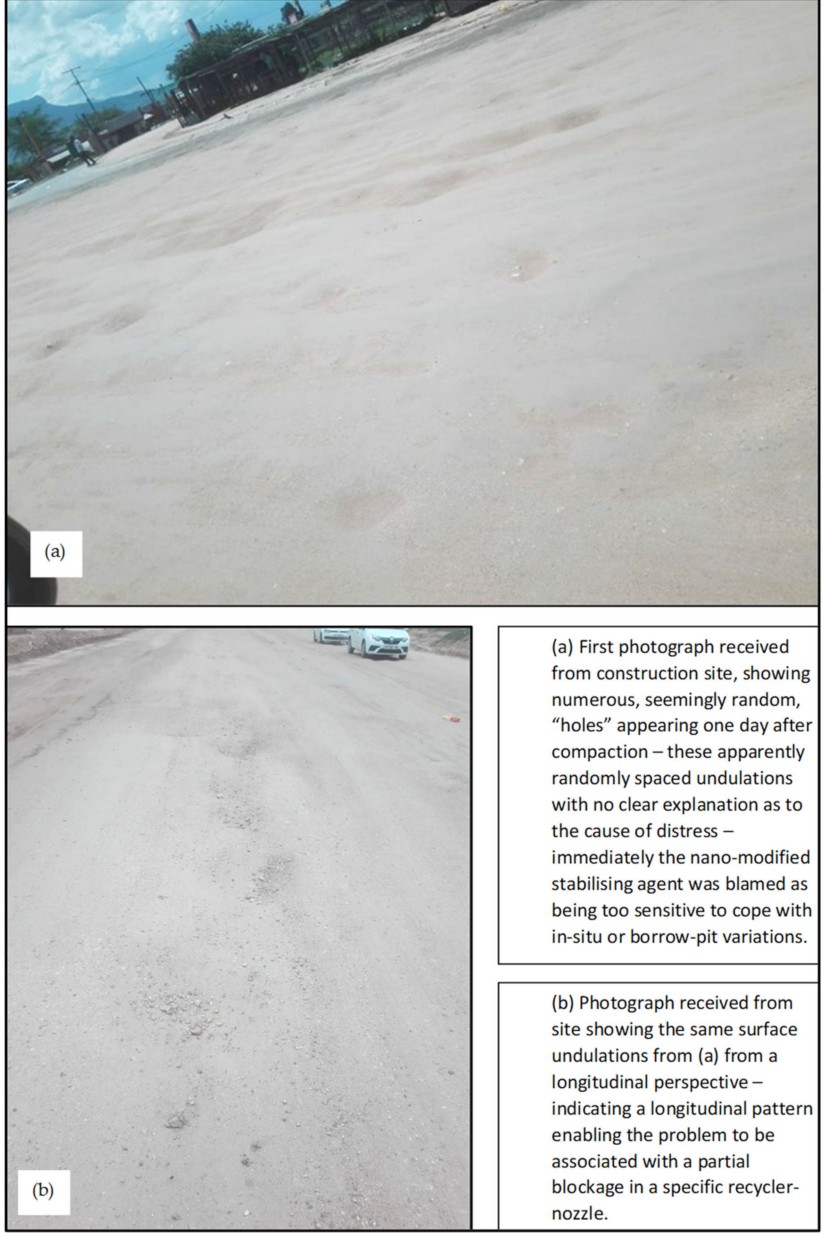

(a) First photograph received from construction site, showing numerous, seemingly random, "holes" appearing one day after compaction – these apparently randomly spaced undulations with no clear explanation as to the cause of distress – immediately the nano-modified stabilising agent was blamed as being too sensitive to cope with in-situ or borrow-pit variations.

(b) Photograph received from site showing the same surface undulations from (a) from a longitudinal perspective – indicating a longitudinal pattern enabling the problem to be associated with a partial blockage in a specific recycler-nozzle.

**Figure 7.** Numerous seemingly random "holes" appearing on the surface, the day after stabilisation and compaction.

**Table 1.** Test results of initial stabilisation and dry reworking of the anionic NME stabilised layer after 24 h.

|  | UCS$_{wet}$ (MPa) | ITS$_{wet}$ (kPa) | Material Classification (Construction—Figure 1) |
|---|---|---|---|
| **Construction specification** | 0.70 | 140 | NME3 |
| **Initial stabilisation** | 2.95 | 232 | NME1 |
| **Dry reworking after 24 h** | 2.40 | 199 | NME2 |
| **Loss in strength measurement** | 19% | 14% |  |

The dry reworking of layers not meeting specifications, or the reworking of NME stabilised pavement layers in future rehabilitation works must be understood in the context of the science behind the anionic NME stabilising agent and the interaction thereof with the granular particles in a pavement layer [19–23]. It is essential to understand that the organofunctional silane modification of the stabilising agent (anionic NME) enables the ability to:

- Cover each particle of the granular material within a pavement layer to become water repellent (hydrophobic) (provided a scientifically based design procedure is followed [8,23]), and
- Form strong chemical bonds between the granular particles and the stabilising agent (i.e., bitumen emulsion or equivalent polymer).

Should a layer need to be reworked for whatever reason, the weakest link will be broken, which is probably either the bitumen stabilising agent (especially when it is relatively freshly stabilised) or, in the case of severely weathered materials, the granular particles break. In the example shown in Figure 6 and Table 1, the primary minerals measured (determined using XRD scans [19]) contained a high percentage of silicon in the form of quarzitic material (40–50%) that is relatively hard [23]. These material particles probably did not break during reworking of the layer with the recycler. The strong chemical bonds between the primary minerals and the nano-silane [22] were also not broken. Hence, the bitumen stabilising agent (still relatively fresh and viscous) sheared during reworking, and then, bound together again during recompacting to generate adequate in-situ strength properties, as confirmed by the UCS and ITS test results. Of course, not all bonds could be recreated, resulting in the measured reduction in UCS and ITS measurements, as shown in Table 1. These results are very specific to the mineralogy of the material and this knowledge, together with the mix results, enabled (empowered) the contractor to rectify the problem at a minimum cost.

The cause of the distress shown in Figure 7 was somewhat more difficult to identify due to the seemingly randomness of the "holes" that became visible on the surface the day after construction, as shown in Figure 7a. However, in a longitudinal direction (Figure 7b), some pattern could be identified, pointing to an irregular supply of the stabilising agent associated with a specific nozzle of the recycler. After inspection, a small stone was found in nozzle No. 6 in the recycler (this stone was most likely picked up with the construction water in a local stream, with no filter having been attached to the pipe used to source the construction water). Initially, in this case, the anionic NME stabilising agent was immediately blamed as not been able to adequately address and stabilise the variation in the properties of the naturally available granular materials used in the construction of the pavement layer.

It is possible that, during construction, design specifications are initially not met for whatever reason, for example, compaction under moisture conditions either too low or too high (this is normally the case with most problems experienced on site – which can often be associated with a lack of adequate experienced supervision) or breakage of construction equipment during the construction process. Experience has shown that in-situ reworking of such a layer, with the addition of 50 per cent of the initial specified anionic NME stabilising agent, ensures that the reworked layer easily surpasses the required specification (i.e., if the initial design required stabilisation of the granular materials was found to be 0.7 per cent anionic NME, reworking with 0.35 per cent anionic NME (approximately 0.2 per cent residual bitumen) will suffice for the reworking of the layer in order to meet the engineering specifications). In these cases, the construction water together with the additional anionic NME stabilising agent will be considerably less than the initial required construction water. During reworking, the water is only used as a distribution agent of the stabilising agent and acts as a lubricant during compaction. Little (if some aggregate brakeage occurs during the remixing) or no absorption of the water by the aggregate (granular particles) occurs and the layer becomes dry at a much-accelerated rate.

These characteristics are demonstrated in Figure 8 [13], showing the change in the characteristics of dolomite material tested before and after stabilisation with a material-compatible NME stabilising agent (in this case the material-compatible NME required a hydroxy conversion treatment (HCT) [20,23] because of the lack of silicon in dolomite). It can be seen that the density in terms of the maximum dry density (MDD) versus moisture content becomes less sensitive to changes in the compaction moisture content as a function of OMC (flatter line). Hence, the required specified density could be achieved at lower moisture conditions. For example, as shown in Figure 8, a density of 97 per cent of the MDD can be achieved at approximately –2.5 per cent of OMC (red arrow to the left of the figure) versus the approximately –0.75 per cent (blue arrow close to the centre of the figure). At a MDD of 2100 kg/m$^3$, the lower required moisture to achieve a density of 97 per cent of MDD could equate to more than 40,000 l/km less water required to compact a 150 mm layer at a width of 7.4 m. This aspect could be a significant factor for the construction of roads where construction water may be a scarcity.

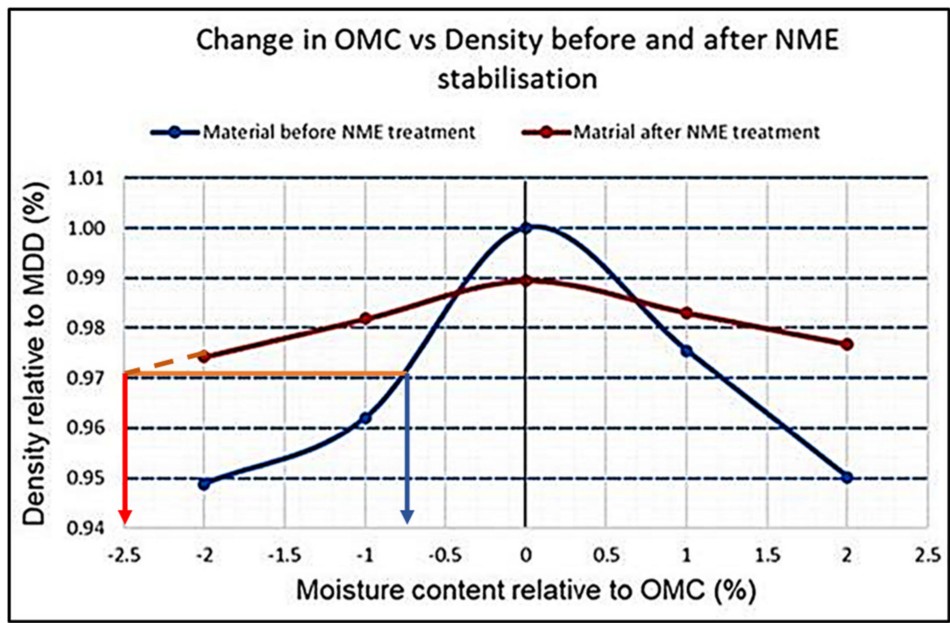

**Figure 8.** Change in material relative density as a function of maximum dry density (MDD) and the change in moisture relative to the OMC before (dark blue) and after (red) stabilisation using a material-compatible anionic NME stabilising agent.

Should the pavement layer require reworking due to any problems (e.g., as shown in Figure 6 or Figure 7) and the design criteria is exceeded by some considerable margin (as indicated in Table 1), with the knowledge of the mineral composition of the granular material available as per recommended test requirements [20,23], dry reworking, as shown in Table 1, could be a viable option, depending on the quality and percentage of the primary mineral present in the granular layer as explained. However, such actions are done solely at the risk of the contractor.

### 4.2.2. Concerns Regarding the Future Rehabilitation of NME Stabilised Pavement Layers

Future rehabilitation of pavement layers originally stabilised with a water-repellent NME stabilising agent is the same as discussed in the previous section. It should be realised that NME technology acts as an aggregate adhesive agent between the granular materials and the stabilising agent with the additional ability to make each granular material particle hydrophobic. The stabilising agent is still an organic substance (bitumen or equivalent polymer of similar characteristics) with the same basic characteristics of normal bitumen stabilisation, which has been used for more than a century in road constructions that have been successfully rehabilitated on numerous occasions. The only change is in the amount

of water required to recompact or restabilise a previously stabilised granular layer using an NME stabilising agent, similar to the graph shown in Figure 8. Less water is required to achieve the specified density, which is a considerable advantage in water-challenged regions of the world. Due to predictions of climate change and earth warming, this factor could become more significant in future road construction projects and could contribute to more sustainable construction materials and methods.

*4.3. Uneven Spray Rates Caused by Blocked Nozzles Noticed of Recycling Equipment*

It is essential that construction be done with sufficient experienced personnel. Figure 9 shows the typical visual appearance of a granular layer that has just been mixed with a NME stabilising agent diluted in construction water. Any blocked nozzles can result in uneven distribution of the stabilising agent within the construction water, as shown in Figure 10. The problem should be immediately detected (provided experienced supervisory personnel are on site) and can be rectified with little additional cost and effort. Blocked nozzles are a common problem often related to poorly cleaned and or maintained equipment.

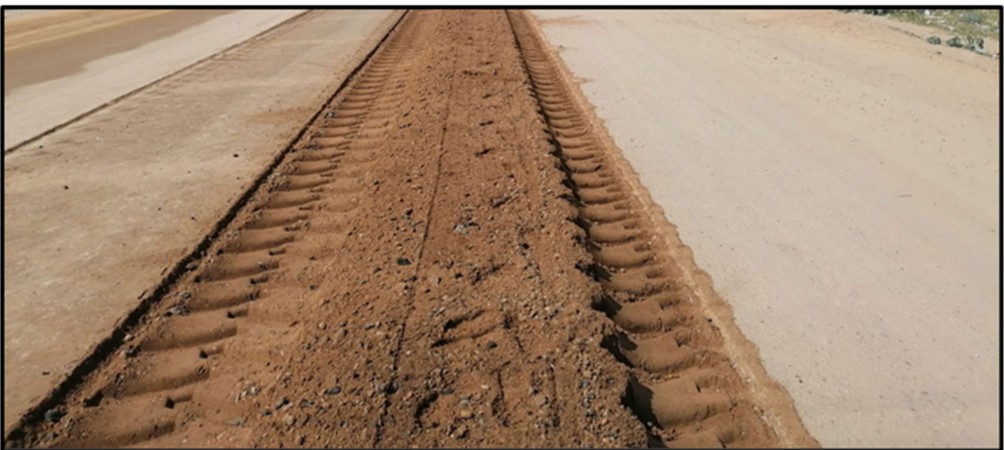

**Figure 9.** Uniform visual appearance of a uniform mix of granular material with an anionic NME stabilising agent in construction water as mixed with a recycler.

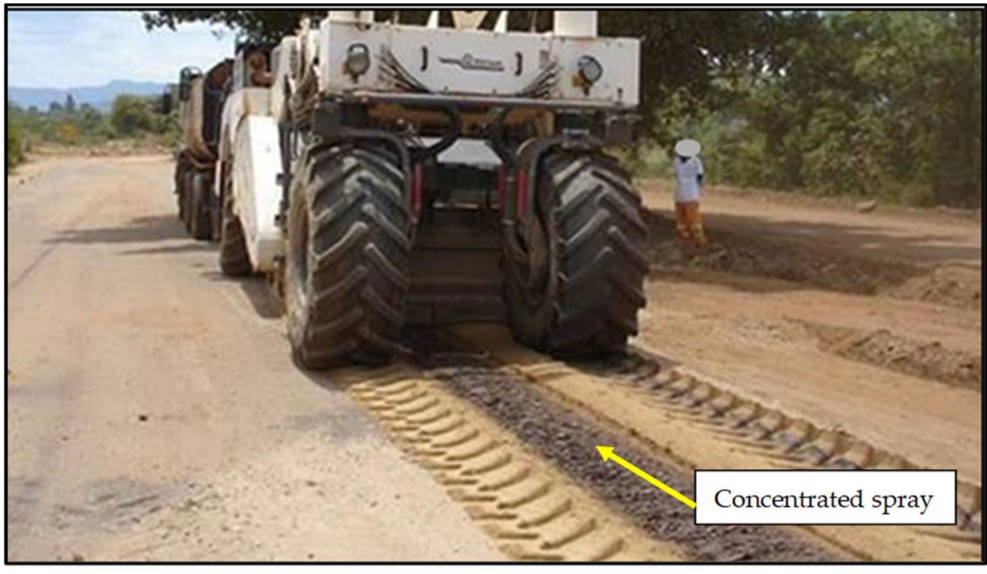

**Figure 10.** Uneven distribution of an anionic NME stabilising agent in construction water behind a recycler, caused by blocked spray nozzles of the recycler.

The NME stabilising agent needs to be evenly distributed throughout the width and depth of any layer during construction to achieve the required engineering properties, similar to any other stabilising agent. Due to the resilient nature of an NME stabilising agent, as discussed in the preceding sections, this problem (Figure 10) can be resolved easily using conventional equipment to evenly distribute and remix the stabilising agent into the layer, as demonstrated in Figure 11.

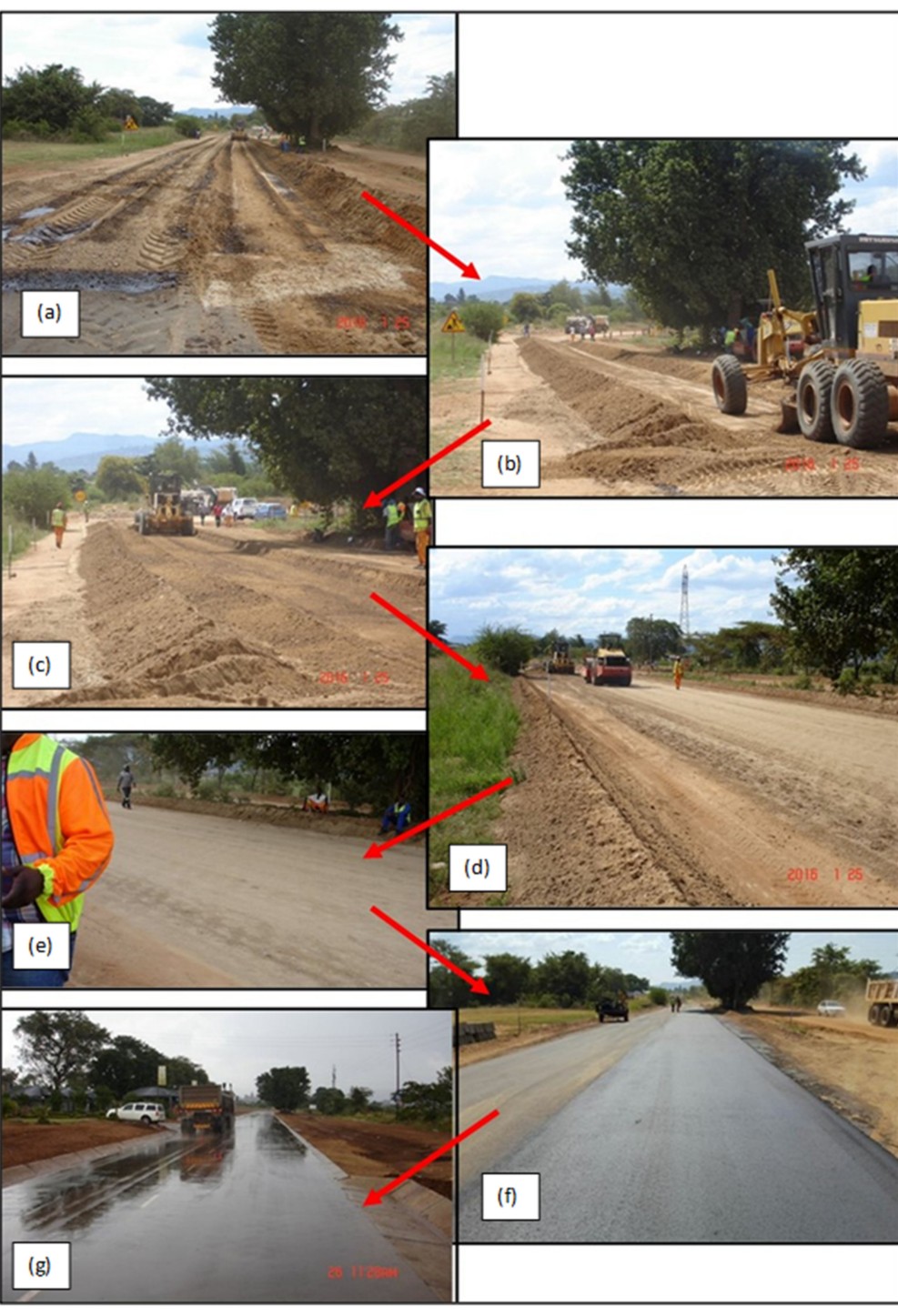

**Figure 11.** Resolving uneven distribution of a NME stabilising agent (Figure 10): (**a–c**) Through the remixing of the road pavement layer using a conventional grader; (**d,e**) compaction of the remixed layer showing the uniformity achieved with the grader; (**f**) primed surfacing of the layer remixed using a grader; (**g**) finished road with surfacing on the section of road with the original uneven distribution of the stabilising agent (Figure 10).

### 4.4. Post-Construction Problems Blamed on the "Nano"-Stabilised Base Layer

In this case, severe rut depth in excess of 50 mm (Figure 12a) crocodile cracking, potholes, and general failed conditions (Figure 12b) were detected at the end of the construction break over the summer holidays, on a section of road constructed just before the summer break. The pavement consisted of 35 mm asphalt surfacing with a 150 mm base layer stabilised with an anionic NME stabilised granular (G7 [27] quality) material. It was immediately argued that the extent of the damage must be related to the "nano" stabilised base layer.

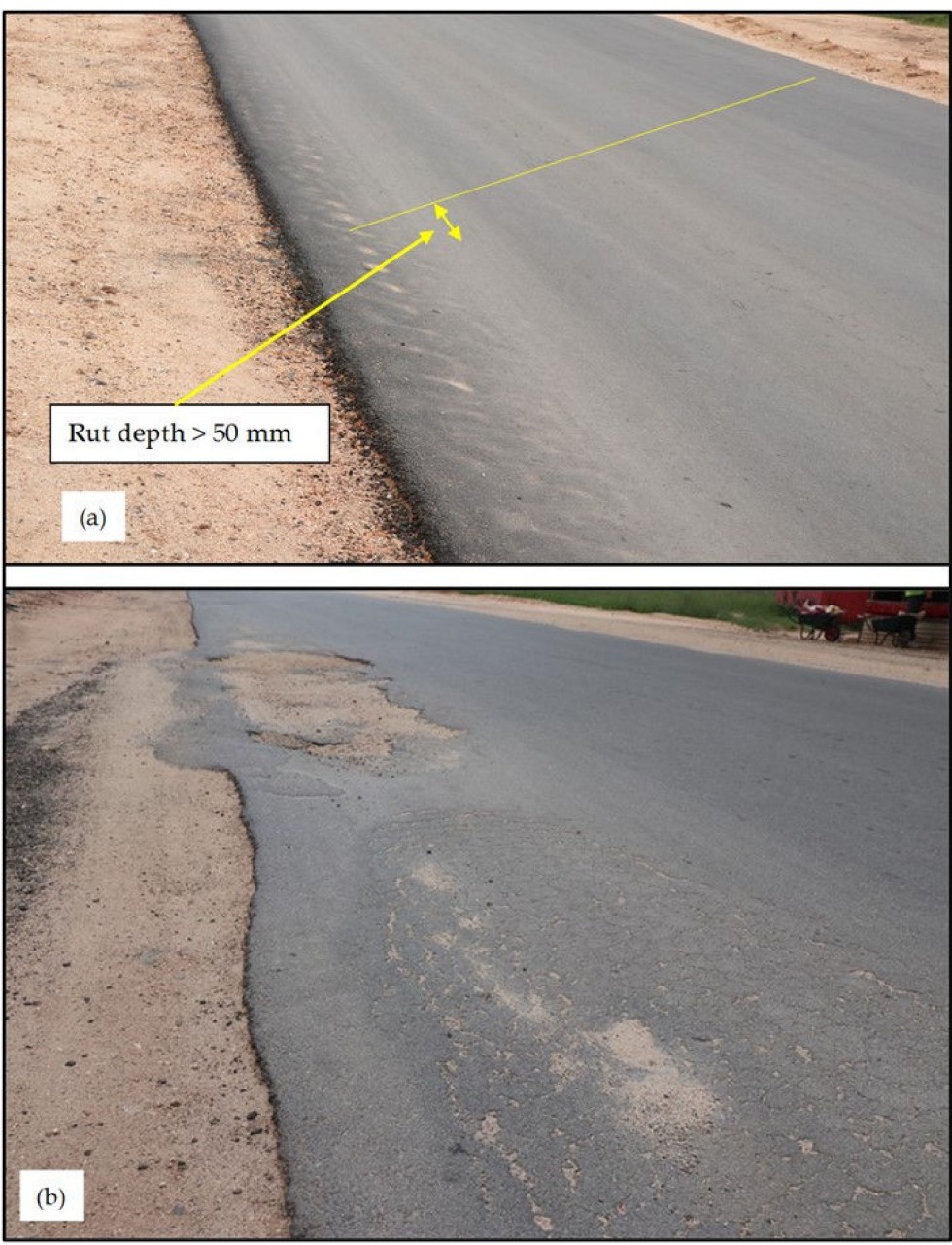

**Figure 12.** Pavement section with 35 mm asphalt surfacing and a 150 mm anionic NME stabilised base layer. Condition of completed section of road one month after the summer break: (**a**) Rut depth in wheel tracks exceeding 50 mm; (**b**) severe distress with crocodile cracking, potholes, and general failures.

A forensic investigation was required to determine the actual cause and mechanism of distress; the evidence could be clearly seen on site by an experienced engineer and a simple

investigation of construction records. An on-site investigation showed that the "crocodile" cracking started as parabolic cracking, indicative of shear during compaction. Similarly, the potholes contained primed sections still attached to the exposed base, indicating that the asphalt surfacing sheared from the primed base with a few potholes exceeding the depth of the surfacing, as shown in Figure 13.

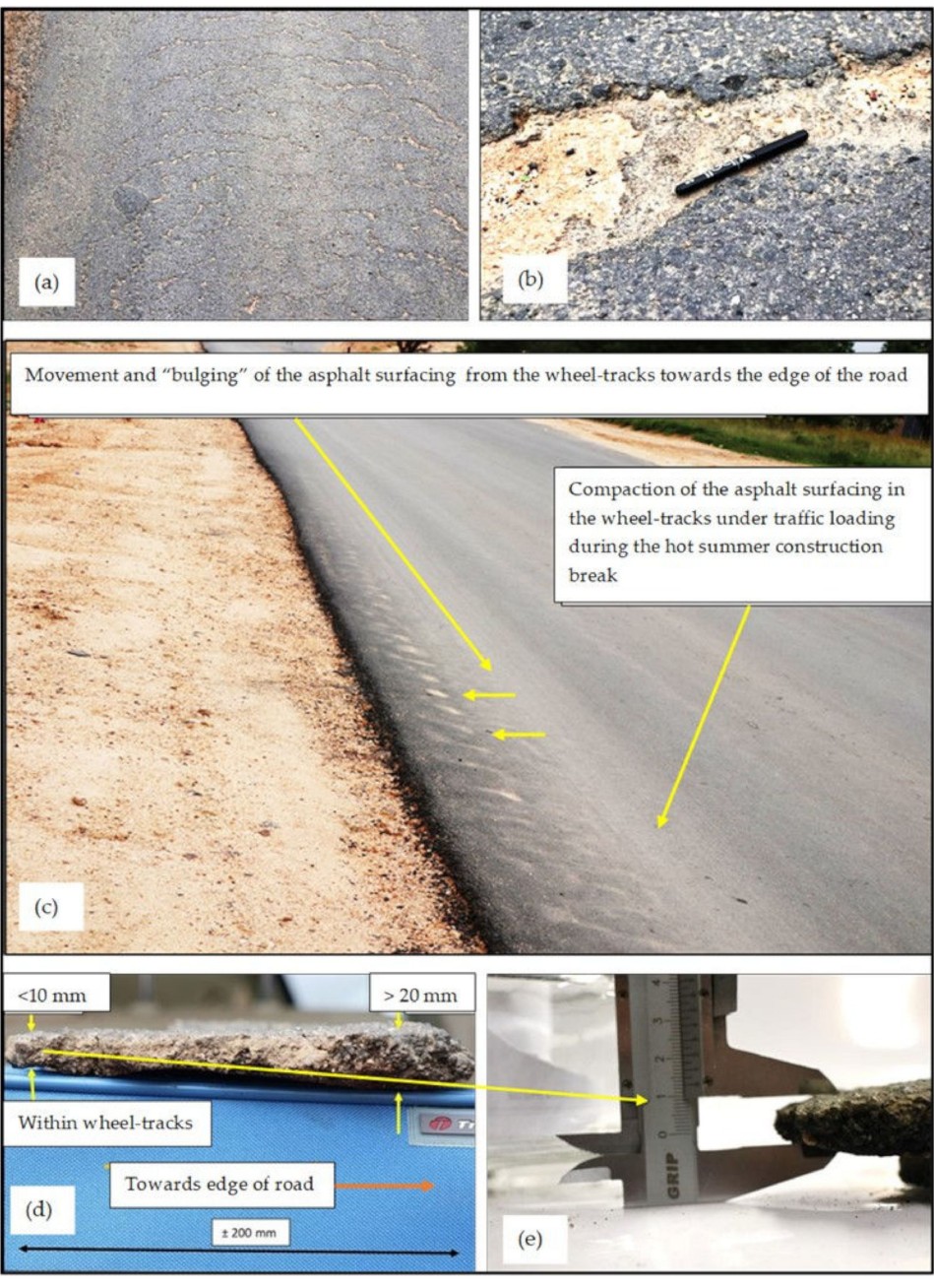

**Figure 13.** (**a**) Parabolic cracking indicative of shear; (**b**) exposed base layer with prime still sticking to the base layer; (**c**) action resulting in rut depth measurements in excess of 50 mm; (**d**,**e**) surfacing taken from road shows a thickness within the wheel tracks of less than 10 mm thickening towards the edge of the road where the thickness exceeded 60 mm.

From the forensic investigation, it was established that, in a haste to finish the section of road before the summer break, the surfacing was placed in rain (specifically prohibited under the project specifications) and compacted under wet and relatively cold conditions. The placement of the surfacing in the rain (without supervision) created a slip plane and lit-

tle adhesion to the base layer. The average compaction measured after construction showed an average compaction density of only 91.5 per cent as compared with the specification of a minimum of 93 per cent. On the hot summer days during the construction break, the asphalt compacted under the action of traffic and moved sideways towards the edge of the road, as shown in Figure 13. The thickness of asphalt examples taken from site indicated a thickness of less than 10 mm in the wheel tracks, resulting into a thickness of more than 60 mm towards the edge of the road. The result was a rut depth of more than 50 mm within the wheel tracks.

The distress, shown and discussed in this example, could easily have been prevented if sound construction practices had been followed. However, the example demonstrates the "first line of attack" that can be used with the introduction of new technologies. It also demonstrates the importance of experienced designers, with necessary practical experience, to be available to construction supervision personnel to give expert advice and the required support to resolve construction problems.

### 4.5. Incompatibility of Materials

A problem generally applicable to pavement engineering is combining incompatible materials in the same design. The same principles apply to the use of NME stabilising agents. These incompatibilities are not normally addressed in design documents and represent a major problem associated with the general use of catalogue designs in combination with documents addressing available stabilising and binder modifications without emphasizing the basic principle of material compatibility. A first rule of thumb borrowed from basic chemistry is "likes prefer likes", i.e., similar binders prefer and work well with the same type of binder. Figure 14 shows the application of two binders on the same anionic NME stabilised sample. On the left, a similar NME modified binder was used in the placement of the chip seal. The chip seal could not be separated from the sample with a trowel resulting in breaking of the stone chips without separating the applied chip seal from the stabilised sample. The binder of the chip seal on the right contained a latex modified binder which showed little adhesion to the anionic NME stabilised sample. In fact, the chip seal on the sample on the right could easily be removed by hand.

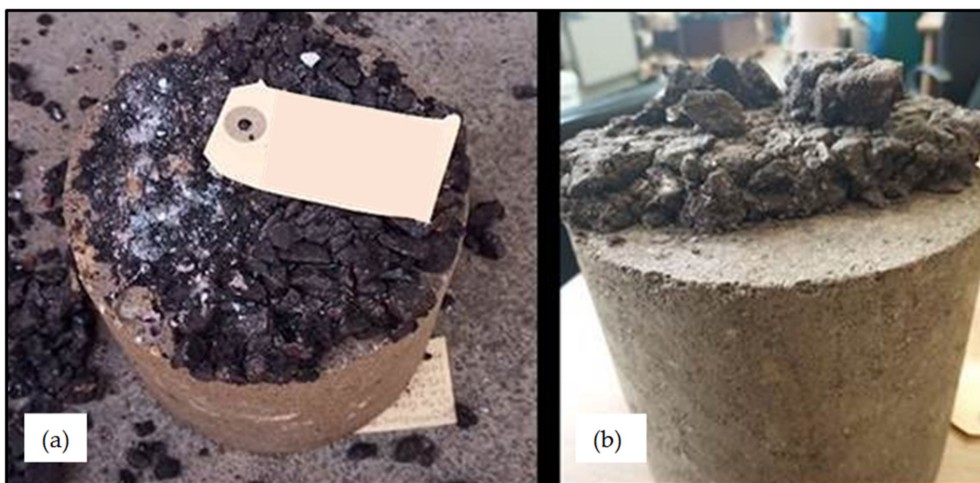

**Figure 14.** Application of different binders to the same anionic NME stabilised samples showing the results using: (**a**) Compatible binders; (**b**) incompatible binders.

The differences in the application of a compatible versus a non-compatible prime to a stabilised base layer are shown in Figure 15. Figure 15a shows the application of a diluted anionic NME prime applied to the surfacing of an anionic NME stabilised sample. Figure 15b shows the consequences of applying a diluted cationic prime to an anionic NME stabilised base layer, resulting in little (or no) adhesion and catastrophic failures.

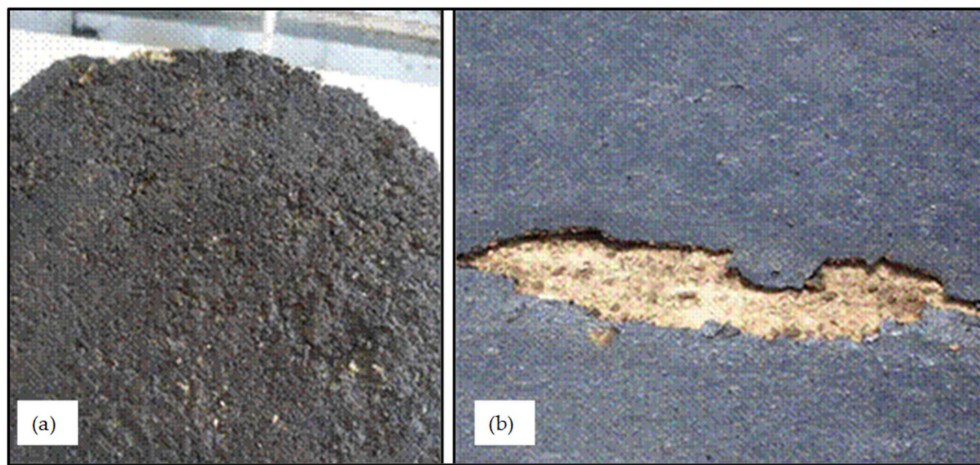

**Figure 15.** (**a**) Material compatible prime of an anionic stabilised sample versus; (**b**) a non-compatible cationic prime applied to an anionic stabilised base layer.

Any selection of a modified binder also needs to be compatible with the future expected performance of a pavement structure. A pavement structure normally requires in the order of two seasons to reach an equilibrium moisture condition. It follows that any surfacing selected, must provide for the pavement structure to expel excess moisture in the form of vapour, independent of the type of modification (or not) used in the construction of the base layer. Unfortunately, many of the newly introduced modifications to bituminous binders "inhibit" the escape of moisture in the form of fumes. Industry documents often include warnings regarding the disadvantages of the use of specific modifications to bituminous materials. However, in practice these warnings are often not taken notice off, with designers only focussing on the possible advantages. It is yet to be seen that researchers confined to laboratories develops test methods that also includes measurements of the "breathability" of modified binders—a practical aspect generally overlooked or just ignored.

The use of a modified binder on a newly constructed road could have severe repercussions, including the following:

- Collection of moisture underneath the surfacing resulting in stripping or detachment of the surfacing from the base, especially in the use of a thin chip seal (Figure 16a,b), or
- Collection of moisture underneath the surfacing in the top of the base layer, resulting in the punching of chips into the base (in the case of a chip seal) with severe bleeding and resultant failure (Figure 16c,d).

*4.6. Addressing and Assuring the Depth of Stabilisation during the Rehabilitation of Roads Utilising In-Situ Materials*

Many premature failures have been investigated and could directly be associated with the depth of reworking and in-situ stabilisation not meeting the specified depths. Modern recyclers are easily programmed to change the depth of reworking which could result in considerable savings both in time and costs of the application of the stabilising agent. Without permanent supervisory staff monitoring every detail, these changes are sometimes difficult to detect. As a result, it is often specified that pavement layers identified to be reworked and stabilised be milled and windrowed to the specified depth. The pavement layer below the milled layer(s) is, then, recompacted before the milled material is reintroduced and stabilised. Using this approach, the designed layer thicknesses are ensured, enabling careful monitoring and quality control as a preventative measure to limit premature failures in a quest for sustainable designs. Figure 17 shows the milling and windrowing of pavement layers before reintroduction and in-situ stabilisation using an anionic NME stabilising agent.

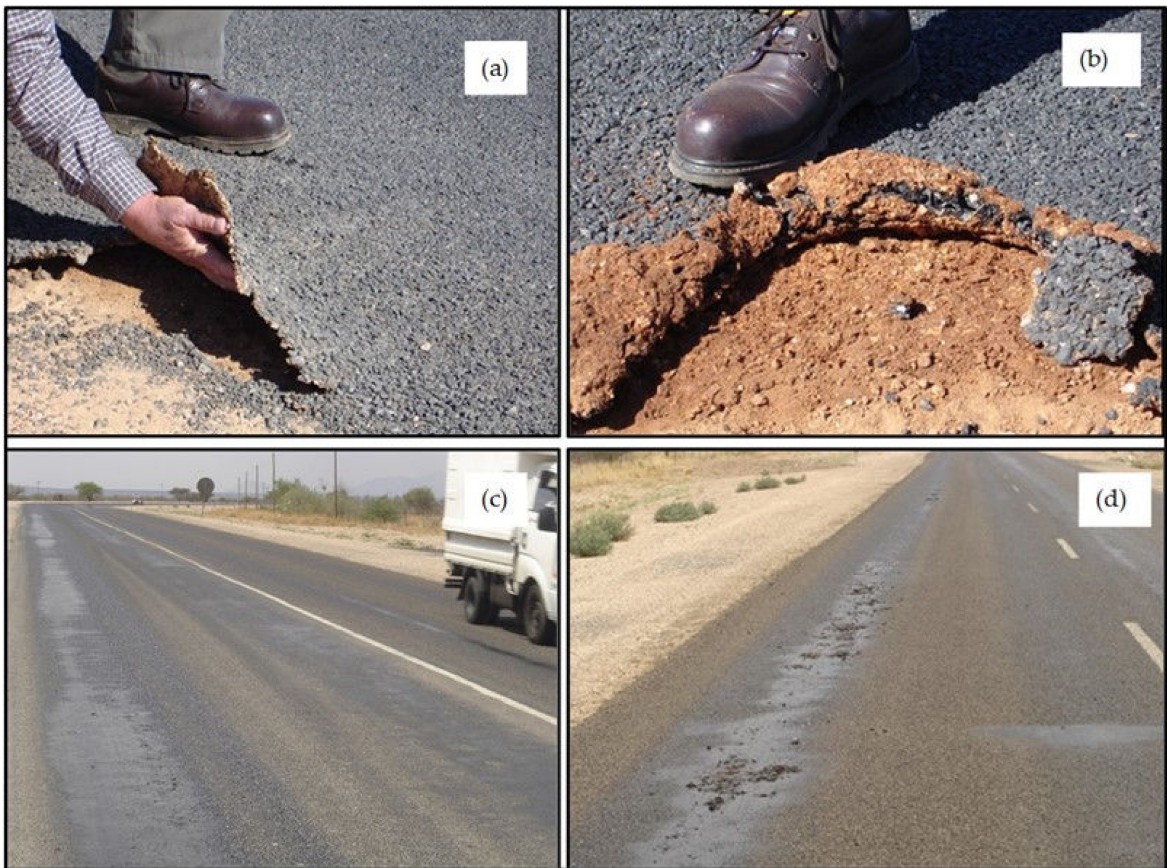

**Figure 16.** (**a**) Detachment of the surfacing from the base layer; (**b**) moisture collecting under a surfacing that inhibits the escape of moisture fumes leading to the conditions shown in (**a**); (**c**) punching of the surfacing into the base layer; (**d**) failure resulting from the punching of the surfacing into the base layer.

Modern construction equipment can easily rework layers up to a depth of at least 450 mm and successfully compact layers of these thicknesses to the specified densities. Before allowing contractors to deviate from the original design by combining layers, these practices should be verified with the design engineer.

The reason for verification of design thicknesses with design engineers before allowing contractors to combine layers using modern construction equipment is simple. The stress/strain distribution of a pavement structure containing, for example, two 150 mm stabilised layers may not necessarily be the same as the stress/strain distribution of a single 300 mm stabilised layer. It follows that the combination of layers within a pavement structure may have an influence on the bearing capacity of the pavement structure. This example is especially relevant for the stabilisation of layers creating a semi-ridged structure with a low tolerance toward flexure (tensile) strength (not applicable to NME-stabilised layers that are flexible, with no cement additives).

Pavement engineering has experienced a strong movement away from empirically based design methods towards Mechanistic-Empirical (ME) design methods over the last few decades. ME design methods, using appliable software, enable a detailed analysis of stress/strain distributions through complex pavement structures. These abilities and associated proven failure theories should be fully utilised by design engineers to approve or disapprove any deviation of the specified design as requested by a contractor. Decisions should not be based solely on the ability of the equipment to perform certain actions and on-site personnel not familiar with the original design approach used. In the case of NME technologies, an increase in layer thicknesses could lead to a longer curing period required

for the layer to expel moisture, especially under relatively cold conditions. In such cases, the use of a surfacing with a modified binder that inhibits the escape of moisture valour from the stabilised pavement layers could prove disastrous, as per previous discussion.

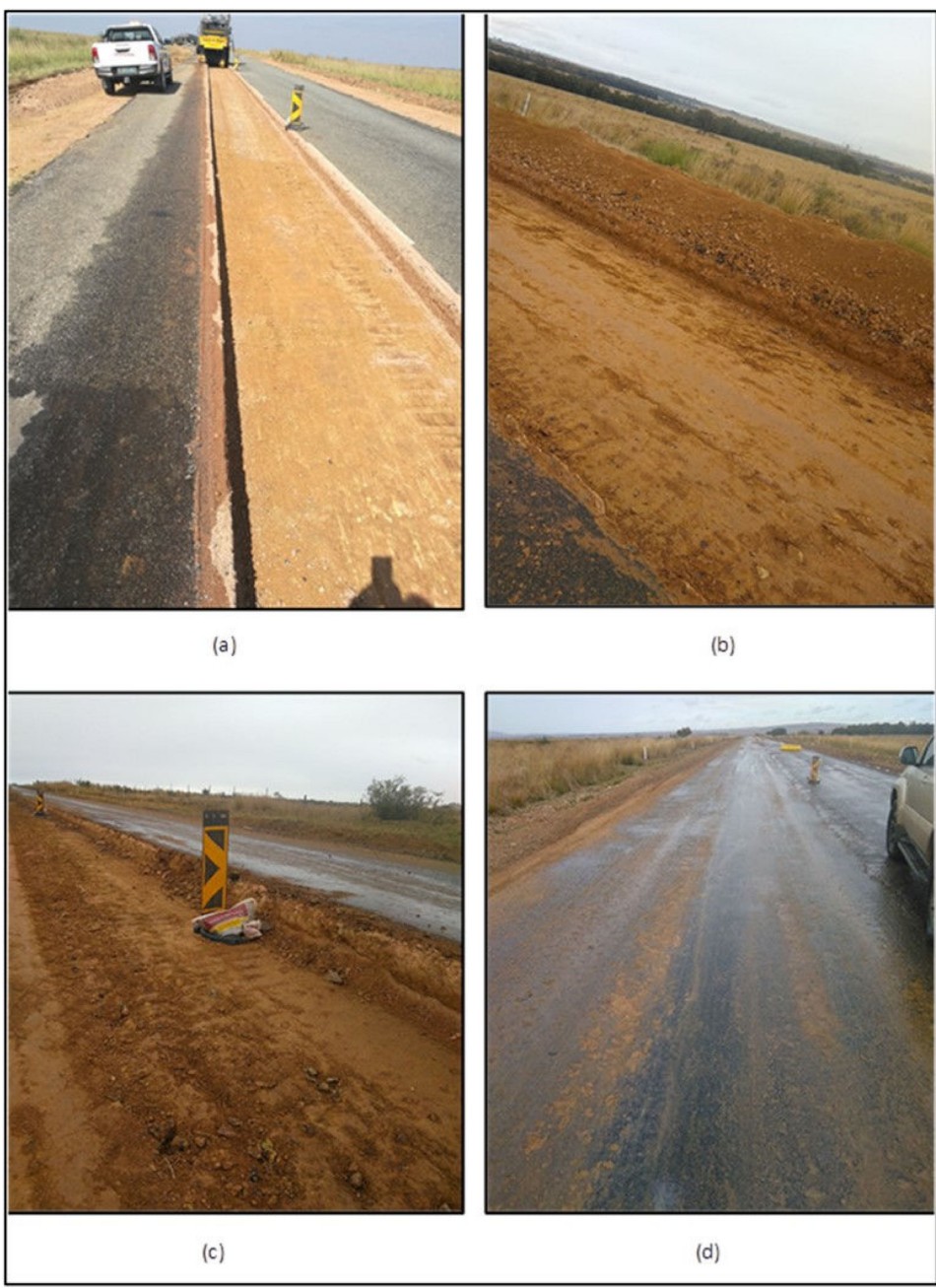

**Figure 17.** (**a**) Milling of existing base and sub-base layers to be stabilised using an anionic NME stabilising agent; (**b**) stockpiling of material next to the road; (**c**) depth of milling and new roadbed to be compacted; (**d**) reintroduced sub-base and base separately stabilised using an anionic stabilising agent.

## 5. Conclusions

Granular materials traditionally classified as marginal or unsuitable for use for in the construction of various roads (as a function of a specific road category) can be stabilised and improved by using material-compatible nanotechnologies in the form of New-age (Nano) Modified Emulsions (NME), which have been tested and evaluated over the years in laboratories through accelerated pavement tests (ATP) and in practice. Scientifically

based materials design methods based on the mineralogy of the granular materials have been developed to ensure that potential risks associated with the introduction of these new technologies are minimised, if not eliminated.

From all the evidence produced, there is little doubt that NME nanotechnology products can contribute considerably towards the construction of roads at considerably reduced unit costs. From previous research, it has been shown that implementing these available and proven technologies can contribute significantly towards the cost-effective delivery of sustainable, much needed road infrastructure without compromising the quality of the end products.

However, the acceptance and rollout of new technologies in the provision of bulk infrastructure, need to be accepted by the construction industry. These nanotechnologies have been shown to be construction friendly, applicable to any existing construction method, and applicable with any available equipment from the most basic to the most advanced. Provided that the recommended materials design method is followed (based on scientific principles of mineralogy and chemistry) and the recommended procurement procedures are followed to ensure that high-quality material-compatible NME stabilising agents are used, it is highly unlikely that NME stabilising will contribute to construction-related problems.

New technologies that are introduced into the construction industry are obvious targets to blame and a "first line of attack" in the case of any construction problems. This article specifically addresses problems encountered on construction sites where anionic NME stabilising agents with marginal or "unsuitable" granular materials have been used in the construction of the base layer. In all cases discussed, the immediate response was to blame the technologies "not proven", relating the use of NME stabilising agents to previous experiences with so-called "snake oils" or "wonder products". However, in all the examples discussed, all problems were related to either procedural, poor supervision, or equipment-related aspects.

The examples discussed in this article aim to counter any uninformed opinions, by identifying the real cause and mechanism of distress, where the natural reaction is to immediately blame the nanotechnology modifications to the stabilising agent. The actual projects covered, using granular materials previously considered to be of unacceptable quality, included roads varying from highways to the upgrade of local access roads and the in-situ rehabilitation of rural and urban streets. Experience has shown that construction-related problems can be prevented and or rectified by following basic sound construction procedures that are applicable to the use of any material, which include the following:

- Use construction water of a specified quality;
- Clean equipment as specified—NME stabilising agents are reactive in nature and react with any residue left in water bowsers, etc. from previous projects, thus, resulting in unusable "blobs" or "strings" of bituminous material;
- Maintain equipment to ensure that equipment-related problems are not the prevailing reason for construction-related problems (most problems encountered could directly be related to problems with the equipment and a general lack of good maintenance policies), and
- Provide supervisory site personnel (foremen) with adequate experience to timeously identify problems, and therefore, rectify these problems in a cost-effective way.

In summary, NME stabilisation of granular materials for the construction of roads has been shown to be forgiving and to be construction friendly, enabling contractors to rectify problems cost-effectively, provided that the cause and mechanism of observed distress can be identified timeously and accurately.

**Author Contributions:** G.J.J. under the directive of the Head of Department of Civil Engineering, W.J.vdM.S., has been leading the research into the provision of affordable road infrastructure at the faculty of Engineering, University of Pretoria. He has been instrumental in the design, implementation, and construction supervision of roads using nanotechnologies. W.J.vdM.S. recognized

the potential of nanotechnology solution in the field of pavement engineering more than a decade ago. G.J.J., through involvement in the private sector, has been responsible for the development of scientific principles, ensuring that implementation can be achieved at a minimum risk. All authors have read and agreed to the published version of the manuscript.

**Funding:** This research received no external funding.

**Institutional Review Board Statement:** Not applicable.

**Informed Consent Statement:** Not applicable.

**Data Availability Statement:** Not applicable.

**Acknowledgments:** The support of GeoNANO Technologies (Pty) Ltd., 18 Davies Road, Wychwood, Germiston, 1401, South Africa, Tel: +27-844078489, www.geonano.co.za, info@geonano.co.za, in support of students in the Department of Civil Engineering, University of Pretoria, Pretoria, South Africa, to test a wide variety of materials as part of final year projects and post-graduate theses, testing the various principles identified in this paper, is acknowledged.

**Conflicts of Interest:** The authors declare no conflict of interest.

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
