# Peer review of "Practical Application of Nanotechnology Solutions in Pavement Engineering: Identifying, Resolving and Preventing the Cause and Mechanism of Observed Distress Encountered in Practice during Construction Using Marginal Materials Stabilised with New-Age (Nano) Modified Emulsions (NME)"

_applsci, doi:10.3390/app12052573_

Round 1
Reviewer 1 Report
Line 45 mentions the beginning of the 2nd millennium (1000-1010 AD) as the time when the potential on nanotechnology applications was recognized in the field of road pavement engineering?
Figures 1 and 2 actually show 2 tables, so they can be converted to the table format accepted by the publication. In the article the authors can describe in more detail the content of these tables. The tables include a limited number of materials, without a breakdown of the mineralogical composition. How the percentages considered by the authors to be normal, both, before and after the application of the stabilization treatment were established?
Why can't the construction of NME Stabilized Granular Pavement Layers be used on more important roads such as highways?
Figure 1 on rows 164 and 166 is Figure 3.
Can only water quality and dirty equipment influence NME Stabilized Granular Pavement Layers? (on section 3).
The acronym OMC used in many places in the article (eg in lines: 218, 220, 221, 316, 339, etc.) must be explained to readers.
The acronym MDD used in fig. 8 and another places from the article must be explained to readers.
An example of a damaged NME is given in the paragraph between lines 276-287. Readers should be given more detailed information regarding the statement "the primary minerals measured contained a high percentage of silicon in the form of quartzitic material that are relatively hard. These material particles did not break during reworking of the layer with the recycler. strong chemical bonds between the primary minerals and the nano-silane were also not broken. Hence, the bitumen stabilizing agent (still relatively fresh and viscous) sheared during reworking and then bound together again during recompacting to generate adequate in-situ strength ". The authors say that these statements are: "properties as confirmed by the test result". What test is about this? In the article is indicated and must be entered more details related to the tests performed, explaining the values ​​obtained. After stabilization with NME, a new material results, with new properties. On this material are required to perform laboratory tests to quantify various properties. Several recipes must be established, and testing must justify an optimal percentage of NME use. What calculation formulas are used? Can the results be verified by a theoretical mathematical calculation?
At lines 319, 320 it is said that the design criteria have been exceeded! What are these design criteria?
On line 323 states that the "dry reworking procedure is not a recommended procedure for remixing". Why does the article say that: it is a viable option? (line 321).
Section before 4.2.2. are not concluded: "It should be realized that the material particles are rendered hydrophobic and that the stabilizing agent is still an organic substance (bitumen or equivalent polymer of similar characteristics)".
The graph shown in Figure 8 does not demonstrate the statement: "Less water will be required to achieve the specified density - a considerable advantage in water-challenged regions of the world."
The execution steps of figure 11 should be presented and explained in more detail.
The statement on lines 471, 472 should be reworded. Does the builder (contractor) not have the legal right to modify the original project, except with the written consent of the designer ?!
The authors state in line with line 505: "In-fact, from the available results, it is evident that the durability of the roads will be improved through the use of material compatible nanotechnologies that introduce water-repellent characteristics which inhibits in-situ chemical weathering of granular materials ". What are these results? Why are these results not included and detailed in the paper?
Why in the paper do the authors not give examples of good practice, where NME was used? Some examples should be given with NME's behaved well over time.
The last 3 conclusions (lines 547, 548, 549) must be justified in comparison with the methods or classical materials used.
Reviewer 2 Report
The paper is a very interesting one, reporting on empirical study into a practical pavement engineering problem. The following are recommended to further improve the paper: 1. NME may be new to many readers, and it will be helpful for the authors to provide some background information about this type of material 2. As an empirical study, the paper should have provided information on the number of construction projects considered. This is absolutely important because the authors tried to explain construction-related problems associated with NME, and it is important to know the number of projects upon which this explanation was based, and whether the conclusion and discussion are valid. 3. On page 3, Lines 103-105, the study's objective is stated. Please this is not clear enough. Make effort to write the objective in clear, straightforward, technical language for readers to understand your intention. 4. On page 2, Lines 81-82, please explain the equation available there. Also, on Line 72, NME appears for the first time, please explain this abbreviation. Also, define OMC on page 8, Line 218. 5. Explain the source of the technical specifications provided in Figures 1 and 2. 6. Figures 1 (c and d) were not seen in the paper (Page 6, Lines 169). 7. Please introduce Figure 3 8. Figure 8 is indistinguishable unless printed in color. Please consider changing the symbols for the series. 9. There were instances of grammatical errors. A thorough check of the write-up will help remedy all these issues. 10. The topic is a bit misleading. The content of the paper is not properly captured by the paper's caption. Please consider revising it. Thanks.Author Response
Please see Attachment

Round 2
Reviewer 1 Report
Comment 1: What does the UCS and ITS tests mean (from line 333). A brief description of the UCS and ITS tests is required. Are the test results close to those expected by calculation? (a percentage comparison is required).
Comment 2: The sentence on lines 544, 545 should be revised.
Comment 3: In the concluding part of the article, the authors must argue in more detail why nanotechnology and especially Modified Emulsions (NME) contribute to increasing the parameters of sustainability (ie the field to which the publication refers).
